# GDF-15 Inhibits ADP-Induced Human Platelet Aggregation through the GFRAL/RET Signaling Complex

**DOI:** 10.3390/biom14010038

**Published:** 2023-12-27

**Authors:** Baikang Xie, Wenjing Tang, Shuang Wen, Fen Chen, Chao Yang, Min Wang, Yong Yang, Wei Liang

**Affiliations:** 1Department of Cardiology, Union Hospital, Tongji Medical College, Huazhong University of Science and Technology, Wuhan 430022, China; bikenxie@hust.edu.cn (B.X.); m202075760@hust.edu.cn (W.T.); m202075817@hust.edu.cn (F.C.); d202281916@hust.edu.cn (M.W.); 2Hubei Key Laboratory of Biological Targeted Therapy, Union Hospital, Tongji Medical College, Huazhong University of Science and Technology, Wuhan 430022, China; 3Hubei Provincial Engineering Research Center of Immunological Diagnosis and Therapy for Cardiovascular Diseases, Union Hospital, Tongji Medical College, Huazhong University of Science and Technology, Wuhan 430022, China; 4Department of Emergency Medicine, Union Hospital, Tongji Medical College, Huazhong University of Science and Technology, Wuhan 430022, China; wenshuang@hust.edu.cn; 5Department of Vascular Surgery, Union Hospital, Tongji Medical College, Huazhong University of Science and Technology, Wuhan 430022, China; ychao@hust.edu.cn

**Keywords:** growth differentiation factor-15, platelet, glial-cell-line-derived neurotrophic factor family receptor α-like, protein kinase B, extracellular signal-regulated kinase

## Abstract

Growth differentiation factor-15 (GDF-15) is proposed to be strongly associated with several cardiovascular diseases, such as heart failure and atherosclerosis. Moreover, some recent studies have reported an association between GDF-15 and platelet activation. In this study, we isolated peripheral blood platelets from healthy volunteers and evaluated the effect of GDF-15 on adenosine diphosphate (ADP)-induced platelet activation using the platelet aggregation assay. Subsequently, we detected the expression of GDF-15-related receptors on platelets, including the epidermal growth factor receptor (EGFR), human epidermal growth factor receptor 2 (HER2), human epidermal growth factor receptor 3 (HER3), transforming growth factor-beta receptor I (TGF-βRI), transforming growth factor-beta receptor II (TGF-βRII), glial-cell-line-derived neurotrophic factor family receptor α-like (GFRAL), and those rearranged during transfection (RET). Then, we screened for GDF-15 receptors using the GDF-15-related receptor microarray comprising these recombinant proteins. We also performed the immunoprecipitation assay to investigate the interaction between GDF-15 and the receptors on platelets. For the further exploration of signaling pathways, we investigated the effects of GDF-15 on the extracellular signal-regulated kinase (ERK), protein kinase B (AKT), and Janus kinase 2 (JAK2) pathways. We also investigated the effects of GDF-15 on the ERK and AKT pathways and platelet aggregation in the presence or absence of RET agonists or inhibition. Our study revealed that GDF-15 can dose-independently inhibit ADP-induced human platelet aggregation and that the binding partner of GDF-15 on platelets is GFRAL. We also found that GDF-15 inhibits ADP-induced AKT and ERK activation in platelets. Meanwhile, our results revealed that the inhibitory effects of GDF-15 can be mediated by the GFRAL/RET complex. These findings reveal the novel inhibitory mechanism of ADP-induced platelet activation by GDF-15.

## 1. Introduction

Activated platelets can contribute to thrombosis, which plays a significant role in many clinical diseases, such as atherothrombosis and deep vein thrombosis [1,2]. Moreover, some antiplatelet activation drugs have been widely used clinically and are crucial in the treatment of thrombotic diseases [3,4]. However, the inhibitory mechanisms of platelet activation have not been fully understood, and there is a need to discover novel targets against platelet activation.

Growth differentiation factor-15 (GDF-15), a member of the transforming growth factor-β (TGF-β) superfamily, acts as a marker of inflammation because it is highly elevated during inflammation to restrain inflammatory response [5]. GDF-15 is also reportedly closely associated with tumor progression, nausea, emesis, and anorexia [6,7,8]. Furthermore, GDF-15 is reportedly strongly correlated with multiple cardiovascular diseases, such as heart failure and atherosclerosis [9,10,11]. Interestingly, our recent study also suggests that plasma GDF-15 is significantly increased in patients with deep vein thrombosis, particularly in those with a more severe thrombus burden [12]. Furthermore, Rossaint et al. reported that GDF-15 can also prevent the activation of platelet integrin αIIbβ3 [13]. In fact, it is similar to the mechanism of the B-type natriuretic peptide (BNP) in heart failure. BNP is secreted mainly by cardiomyocytes that react to the volume overload of the ventricle. Increased BNP indicates elevated severity and a worse prognosis of heart failure. At the same time, treating with recombinant human BNP can significantly improve the condition of patients with heart failure [14]. According to previous studies, GDF-15 is abundantly expressed in the macrophages and endothelial cells [15,16]. Endothelial injury and inflammation are common pathways through which the risk factors trigger platelet activation and thrombus formation. Therefore, vascular endothelial cell injury and inflammation promote prothrombotic alterations, which induce increased GDF-15 secretion and exert its anti-inflammatory and anti-platelet effects.

ADP is a second-wave mediator, which is released following GPVI activation and contributes to platelet activation via the stimulation of various platelet receptors [3]. The binding of ADP to the platelet receptors P2Y1 or P2Y12 leads to a platelet shape change, calcium mobilization, and, finally, platelet aggregation [17]. Thus, platelet aggregation is the result of ADP-induced platelet activation, and conversely, platelet aggregation can reflect platelet activation. Our previous study indicated that GDF-15 seems to inhibit platelet aggregation through the ADP-P2Y1/P2Y12 pathway but not the collagen-GPVI/integrin α2β1 pathway [12]. In this study, we mainly focused on ADP-induced platelet aggregation and the mechanism by which GDF-15 affects ADP-induced platelet activation and aggregation.

Although the adhesion receptor αIIbβ3 is essential for platelet aggregation and thrombosis, it does not seem to be the direct target receptor for GDF-15 [13,18,19]. Thus, the mechanism underlying the effect of GDF-15 on platelet function needs to be further clarified. In this study, we isolated platelets from the peripheral blood of healthy volunteers and investigated the role and mechanism of GDF-15 in the regulation of ADP-induced platelet activation.

## 2. Materials and Methods

### 2.1. Preparation of Human Platelets, Erythrocytes, and Leukocytes

As previously described [20,21], blood samples were obtained from the antecubital venipuncture of healthy volunteers. The samples were collected into a 2 mL vacutainer tube containing 3.2% sodium citrate. Then, the platelet-rich plasma (PRP) was obtained via the centrifugation of the sample at 100× *g* for 10 min, and the plasma was further centrifuged again at 2000× *g* for 15 min to obtain platelet-poor plasma (PPP). Platelets are obtained from PRP and washed twice via centrifugation at 600× *g*. Erythrocytes were collected from the lower layer after the first centrifugation, and leukocytes were collected from the middle layer. The erythrocyte lysis buffer was added to the leukocyte. Next, the plasma containing the erythrocyte lysis buffer was further centrifuged at 2000× *g* for 5 min to obtain the leukocytes. Platelets, erythrocytes, and leukocytes were suspended with the HEPES-Tyrode buffer (145 mM NaCl, 1 mM MgSO_4_, 5 mM KCl, 10 mM HEPES, 0.5 mM Na_2_HPO_4_, and 5 mM D-glucose; PH 6.5) supplemented with 100 nM prostaglandin E1 and washed twice at 600× *g* for 10 min each. Finally, the washed platelets, erythrocytes, and leukocytes were suspended in the HEPES-Tyrode buffer. The platelet count was adjusted to 10^8^/mL. All human blood samples were obtained with informed consent from all the participants. The study was approved by the Ethics Committee of Union Hospital, Tongji Medical College at Huazhong University of Science and Technology ((2022) 0124-01) in compliance with the Declaration of Helsinki.

### 2.2. Platelet Aggregation Assay

Platelet aggregation was detected using a turbidimetric aggregation monitoring device (AggRAM, Helena Laboratories, Beaumont, TX, USA) [20,21]. First, PPP and then PRP were obtained as described above. PRP was pre-warmed to 37 °C in a resting state. PPP was used as a reference to indicate 100% aggregation. The platelet aggregation assay was performed as soon as possible after platelet acquisition. Subsequently, PRP was stimulated with 5-μM ADP (Helena Laboratories, USA) for 5 min to induce platelet aggregation. The experimental group samples were separately incubated with different concentrations of GDF-15 (10 ng/mL, 20 ng/mL, and 40 ng/mL) (Peprotech, Cranbury, NJ, USA) for 15 min before ADP stimulation. The RET agonist BT-13 (HY-124401, MCE) or RET inhibitor SPP86 (HY-110193, MCE) were used to explore the mechanism of inhibition of GDF-15 on ADP-induced platelet aggregation. Finally, platelet aggregability was evaluated by the maximal percentage of platelet aggregation.

### 2.3. GDF-15-Related Receptors Microarray

According to a previously described method [22], the GDF-15-related receptor microarray comprises five recombinant human proteins, including those rearranged during transfection (RET, 11997-H08H1, Sino Biological, Beijing, China), the epidermal growth factor receptor (EGFR, 10001-H08S, Sino Biological, Beijing, China), human epidermal growth factor receptor 3 (HER3, 10201-HCCH, Sino Biological, Beijing, China), TGF-βRI (10459-H02H, Sino Biological, Beijing, China), and GFRAL (9467-GR, R&D Systems, Minneapolis, MN, USA). Each of these proteins was printed in duplicate. Meanwhile, GDF-15 (Peprotech, USA) was biotinylated using an antibody array assay kit (Full Moon Biosystems, Sunnyvale, CA, USA) and diluted to different concentration gradients of 0.1 μg/mL, 1.0 μg/mL, and 10 μg/mL. Then, the microarray was blocked with a blocking buffer (3% BSA in 0.1% (*v*/*v*) PBS plus 0.1% (*v*/*v*) Tween 20_ for 1 h at room temperature. The microarray was incubated with the diluted GDF-15 for 2 h at room temperature. After washing with Tris-buffered saline with 0.1% Tween^®^ 20 detergent three times for 5 min each, the microarray was incubated with Cy3–streptavidin (Thermo Fisher Scientific, Waltham, MA, USA) for 1 h at room temperature and washed three times again for 5 min each. The microarray was then centrifuged for 5 min at 100× *g* to remove the double-distilled water, followed by scanning using a GenePix 4000B (Axon Instruments, Sunnyvale, CA, USA) for visualization. 

### 2.4. Western Blot

Human platelets, erythrocytes, and leukocytes were obtained as described above. According to the purpose of the experiment, platelets were treated with GDF-15 in the presence or absence of the RET agonist BT-13 (HY-124401, MCE) or RET inhibitor SPP86 (HY-110193, MCE). Washed cells were lysed in a protein lysis buffer with protease and phosphatase inhibitor cocktails (HY-K0010, MCE) on ice for 30 min, followed by centrifugation at 12,000 rpm for 15 min and denaturation for 10 min. Then, protein samples (20 μg) were separated by the gradient 7.5–12.5% sodium dodecyl sulfate-polyacrylamide gel electrophoresis and transferred to nitrocellulose membranes. The membranes were blocked with 5% fat-free dry milk in Tris-buffered saline with 0.1% Tween 20 for 2 h at room temperature and incubated overnight at 4 °C with various primary antibodies, including GFRAL (ab235111, 1:2500, Abcam, Waltham, MA, USA), RET (ab134100, 1:1000, Abcam, Waltham, MA, USA), TGF-βRI (YT4627, 1:1000, Immunoway, Plano, TX, USA), TGF-βRII (ab259360, 1:1000, Abcam, Waltham, MA, USA), EGFR (YT1479, 1:1000, Immunoway, Plano, TX, USA), HER2 (YM3045, 1:1000, Immunoway, Plano, TX, USA), HER3 (YT1608, 1:1000, Immunoway, Plano, TX, USA), phospho-AKT (#4060, 1:2000, CST, Danvers, MA, USA), AKT (#4685, 1:1000, CST, Danvers, MA, USA), phospho-ERK1/2 (#4370, 1:2000, CST, Danvers, MA, USA), ERK1/2 (#4695, 1:2000, CST, Danvers, MA, USA), Janus kinase 2 (JAK2; ab108596, 1:1000, Abcam, Waltham, MA, USA), and phospho-JAK2 (#3771, 1:1000, CST, Danvers, MA, USA), followed by incubation with the horseradish peroxidase-conjugated secondary anti-rabbit (ANT020, AntGene, Wuhan, China) or anti-mouse (ANT022, AntGene, Wuhan, China) antibody at room temperature for 2 h. After the visualization and semi-quantification of target bands, the membranes were stripped and incubated with anti-GAPDH (60004-1, 1:3000, Proteintech, Wuhan, China) to ensure equal loading.

### 2.5. Immunoprecipitation Assay

According to the method described above, the washed platelets were obtained and resuspended in the HEPES-Tyrode buffer pre-warmed to 37 °C. The platelet suspension was successively incubated with 40 ng/mL of GDF-15 for 20 min at 37 °C and lysed with equal volumes of the 2 × NP-40 lysis buffer (300 mm NaCl, 100 mm Tris-HCl, 2% NP-40, 2 mm NaF, and 2 mm EDTA) containing protease and phosphatase inhibitors on ice for 30 min. The supernatant was collected after centrifugation at 12,000 rpm for 15 min, followed by incubation with the anti-GDF-15 antibody (ab206414, Abcam) or the isotype IgG control (ab172730, 7ed to each sample and incubated for 2 h at 4 °C. The precipitates were separated by centrifugation at 3000 rpm for 3 min and rinsed three times with the NP40 lysis buffer. Finally, the washed immunoprecipitated beads were boiled in loading buffers for further Western blot analysis. VeriBlot for the IP detection reagent (ab131366, 1:3000, Abcam), which only recognizes native (non-reduced) antibodies, was used for immunoblotting.

### 2.6. Statistical Analysis

Data are shown as the means ± SEM. Data normality was assessed using the Shapiro–Wilk test. Differences between the groups were evaluated via the one-way analysis of variance followed by Tukey’s multiple comparisons test (for homogeneous variance) or the Tamhane test (for nonhomogeneous variance). The values of *p* < 0.05 were considered statistically significant. All analyses were performed using SPSS version 22.0 (Chicago, IL, USA). 

## 3. Results

### 3.1. GDF-15 Inhibits Human Platelet Aggregation Triggered by ADP

To determine whether GDF-15 has a role in inhibiting platelet activation, we first investigated its effects on platelet aggregation. We found that platelets treated with both GDF-15 and ADP exhibited markedly impaired platelet aggregation compared to those incubated with ADP alone. Furthermore, GDF-15 reduced platelet aggregation in a dose-dependent manner (*p* < 0.05; Figure 1A,B).

### 3.2. Detecting the Expression of Receptors of GDF-15 in Human Platelets

By reviewing the related literature, we found that the epidermal growth factor receptor (EGFR), human epidermal growth factor receptor 2 (HER2), human epidermal growth factor receptor 3 (HER3), transforming growth factor-beta receptor I (TGF-βRI), transforming growth factor-beta receptor II (TGF-βRII), glial-cell-line-derived neurotrophic factor family receptor α-like (GFRAL), and those rearranged during transfection (RET) may be associated with the effect of GDF-15 on platelets. Thus, we immunoblotted human platelet lysates for these receptors. The results revealed that GFRAL, RET, EGFR, HER3, and TGF-βRI are expressed on human platelets (Figure 2A–C). However, we did not observe the expression of HER2 and TGF-βRII on human platelets (Figure 2B,C). In addition, we also conducted additional experiments to validate the WB using positive controls for anti-TGFbetaR II and HER2 (Appendix A).

### 3.3. GDF-15 Interacts with GFRAL in Platelets 

Since GFRAL, RET, TGF-βRI, EGFR, and HER3 are expressed on platelets, we determined to identify the binding partner of GDF-15 among these molecules. Thus, we used their recombinants and further developed a GDF-15-related receptor microarray in vitro, the results of which indicate that GFRAL is the only molecule showing high-affinity binding with GDF-15 (Figure 3A). To further verify whether there was an interaction between GDF-15 and endogenous GFRAL on the platelets, co-immunoprecipitation was performed. The results showed that GDF-15 can interact with endogenous GFRAL on platelets as GDF-15 can pull down endogenous GFRAL (Figure 3B).

### 3.4. GDF-15 Inhibits ADP-Induced AKT and ERK Activation

To elucidate the signaling pathways associated with the effect of GDF-15 on platelet activation, we evaluated the effect of GDF-15 on the activation of the extracellular signal-regulated kinase (ERK), protein kinase B (AKT), and Janus kinase 2 (JAK2) pathways. Compared to resting platelets, platelets pretreated with ADP exhibited a notable elevation in the phosphorylation level of ERK, AKT, and JAK2 (Figure 4A–C). However, the administration of GDF-15 substantially downregulated the level of phospho-ERK and phospho-AKT but not of phospho-JAK2 (Figure 4A–C).

### 3.5. GDF-15 Inhibits ADP-Induced AKT and ERK Activation through GFRAL/RET Signaling Complex

To verify that GDF-15 acts through the GFRAL/RET receptor complex, we carried out further experimental exploration. Similar to other members of the GDNF family of receptors, GFRAL was thought to rely on its coreceptor RET to exert biological effects because it failed to induce intracellular signaling by itself. Thus, platelets were incubated with RET agonists (BT-13) or RET inhibitors (SPP-86) prior to the GDF-15 intervention. The levels of P-AKT and P-ERK were significantly increased after the stimulation with 5 μM of ADP (*p* < 0.01, Figure 5A–C). Compared to the ADP group, pretreatment with 40 ng/mL of GDF-15 for 15 min before ADP stimulation significantly decreased the levels of P-AKT (*p* < 0.01, Figure 5A–C) and P-ERK (*p* < 0.05, Figure 5A–C). More importantly, we found that the activation of RET with BT-13 negated the effect of GDF-15 on P-AKT and P-ERK (*p* < 0.05, Figure 5A–C), while the inhibition of RET with SPP-86 enhanced the inhibitory effects of GDF-15 on the activation of the AKT and ERK pathways (*p* < 0.05, Figure 5A–C).

### 3.6. GDF-15 Inhibits ADP-Induced Human Platelet Aggregation through GFRAL/RET Signaling Complex

We conducted additional experiments to demonstrate how the effect of GDF-15 on platelet aggregation is relative to the GFRAL/RET signaling complex using RET agonists (BT-13) or RET inhibitors (SPP-86). Compared with unstimulated resting platelets, ADP significantly induces platelet aggregation (*p* < 0.0001, Figure 6A,C). Compared to the ADP group, pretreatment with 20 ng/mL GDF-15 for 15 min before the ADP stimulation significantly decreased the ratio of platelet aggregation (*p* < 0.01, Figure 6B,C). More importantly, we found that the activation of RET with BT-13 negated the effect of GDF-15 on platelet aggregation (*p* < 0.05, Figure 6B,C), while the inhibition of RET with SPP-86 enhanced the inhibitory effects of GDF-15 on platelet aggregation (*p* < 0.05, Figure 6B,C).

## 4. Discussion

The mechanism of the GDF-15-mediated inhibition of platelet activation has not been adequately understood, particularly because the receptor of GDF-15 is not fully clear. ADP is an important platelet agonist and plays an important role in platelet activation [3]. And the antagonist of the ADP receptor P2Y12 is a key drug target for platelet activation and thrombotic diseases [3]. Moreover, previous studies have confirmed that ADP binds to two platelet purinergic receptors, P2Y1 and P2Y12, to induce platelet aggregation by suppressing the cAMP signaling pathway [23]. GDF-15 was reported to promote EGFR signaling, which could enhance the activation of cAMP [24,25]. Therefore, we speculated that GDF-15 may affect ADP-induced platelet activation. Thus, we examined the effect of GDF-15 on ADP-induced platelet aggregation. The results show how GDF-15 inhibited dose-dependently of ADP-induced platelet aggregation.

In recent years, the identification of the GDF-15 receptor has attracted increasing interest among researchers. Currently, the potential receptors of GDF-15 can be divided into three classes. The first sort of receptor is the transforming growth factor-beta receptor. GDF-15 was proposed to promote esophageal squamous cell carcinoma progression via the TGF-βRII signaling pathway [26]. In addition, both TGF-βRI and TGF-βRII were reported to interact with GDF-15 in DC cells as essential for the GDF-15-induced inhibition of neutrophil activation [27,28]. Our results show that TGF-βRI and not TGF-βRII are expressed on human platelets. The latter focuses on the human EGFR family, which comprises EGFR, HER2, HER3, and HER4. Both Xu et al. and Carrillo et al. observed that the effects of GDF-15 involve EGFR signaling [24,29]. Moreover, Li et al. revealed that GDF-15 promoted the proliferation of cervical cancer cells through the HER2 receptor [7]. In addition, Park et al. also found that GDF-15 could induce the phosphorylation of EGFR, HER2, and HER3 [30]. Thus, EGFR, HER2, and HER3 may be the receptors of GDF-15. Next, we examined their expression in platelets and observed that EGFR and HER3 are distributed on human platelets, but HER2 is not. The third candidate is GFRAL, which is required for the anti-obesity effects of GDF-15 [31]. Since RET functions as a coreceptor with GFRAL and is essential for GDF-15-mediated signaling [32], we also examined the expression of RET in addition to GFRAL on platelets. Interestingly, both GFRAL and RET were observed to exist on human platelets. Therefore, these candidates—GFRAL, RET, EGFR, HER3, and TGF-βRI—may be the binding partners of GDF-15 on platelets. To further identify which one of these receptors is expressed on the platelets that GDF-15 can bind with, we performed the GDF-15-related receptors microarray in vitro. Among these recombinant proteins, as the results showed, GFRAL was the only specific protein that GDF-15 could bind with. Given the significant differences between the external and internal environments, we further used the immunoprecipitation assay to confirm whether GDF-15 can also interact with the GFRAL expressed on platelets. The results revealed that GDF-15 can pull down endogenous GFRAL expressed on human platelets.

GFRAL, identified as a distant homolog of the glial-cell-line-derived neurotrophic factor (GDNF) family of receptors, reportedly plays a significant role in neuronal cell survival [33]. GFRAL was initially thought to be expressed exclusively in brain tissues; however, some subsequent studies revealed that it is also expressed in pancreatic cancer cells, prostate cancer cells, and vascular endothelial cells [34,35,36]. Recently, GFRAL was identified as a high-affinity receptor of GDF-15, which is associated with metabolic effects [31,32,37,38]. Similar to other members of the GDNF family of receptors, GFRAL is thought to rely on its coreceptor RET to exert biological effects because it failed to induce intracellular signaling by itself [31,32,37,38]. In this current study, we observed that GFRAL is expressed on human platelets and that it can bind to GDF-15 in vitro. Furthermore, we also found that RET, the coreceptor of GFRAL, was distributed on human platelets. Thus, GFRAL on platelets may be the critical target for GDF-15.

Among the signaling pathways, ERK, AKT, and PLC-γ pathways were recently revealed to be the downstream signaling pathways of GDF-15/GFRAL [31,32,37,38,39]. Protein kinase B (AKT), an important substrate of phosphatidylinositol-3-kinase (PI3K), was associated with platelet activation [40]. The ERK pathway and JAK pathway also played important roles in platelet activation [41,42]. In platelets, PLC-γ signaling was predominantly associated with platelet activation mediated by glycoprotein VI (GPVI) and integrin α2β1 [43], which are two major collagen receptors. Our recent study shows that GDF-15 does not inhibit collagen-induced platelet aggregation [12], indicating that the PLC-γ pathway may not be essential for the inhibitory effect of GDF-15 on platelets. Thus, we evaluated the effects of GDF-15 on the activation of AKT, ERK, and JAK2, respectively. Consequently, we found that the phosphorylation levels of ERK and AKT were dramatically decreased in platelets treated with GDF-15 and ADP when compared to patients stimulated with ADP alone. However, the level of phospho-JAK2 in the ADP and GDF-15 group did not show a significant difference compared to that in the ADP alone group. Contrary to our results, Yang et al. determined that GDF-15 could lead to a higher level of the phosphorylation of AKT and ERK in HEK cells co-transfected with hGFRAL and hRET [31]. This could be related to different cell types in in vitro environments. GDF-15 has been reported to inhibit the phosphorylation of ERK and AKT, probably through CD48, TGF-βRII, and EGFR [29,44,45]. However, we and Sugiyama et al., respectively, confirmed that normal human peripheral blood platelets do not express TGF-βRII and CD48 [46]. Although we detected the expression of EGFR on platelets, our GDF-15-related receptor microarray assay did not detect the interaction between GDF-15 and EGFR. Thus, our discovery that GDF-15 inhibits ADP-induced AKT and ERK activation in platelets complementarily demonstrates the novel role of platelet GFRAL in the inhibition of AKT and ERK activation and platelet function. Therefore, we speculate that GDF-15 could inhibit ADP-induced platelet aggregation by inhibiting ADP-induced AKT and ERK pathway activation through the GFRAL/RET complex.

To verify that GDF-15 acts via the GFRAL/RET receptor complex, we preincubated platelets with the RET agonist or inhibitor prior to intervention with GDF-15 and found that the RET agonist counteracted the inhibitory effect of GDF-15 on ADP-induced AKT and ERK pathways activation, while the inhibition of RET enhanced the inhibitory effect of GDF-15 on ADP-induced AKT and ERK pathways activation. Therefore, GDF15 may competitively bind to the GFRAL/RET receptor complex, leading to the inhibition and activation of downstream AKT and ERK pathways. GSK-3β and PDE3A have been reported to be the possible main effectors of AKT in platelets [47,48,49]. As reported previously [50], cPLA2 may serve as the important downstream target of ERK. Thus, GDF-15 may act on GSK-3β, PDE3A, and cPLA2 by inhibiting AKT and ERK activation, eventually leading to the inhibition of integrin αIIbβ3 activation, which is vital for platelet function and thrombosis [51].

Taken together, to our knowledge, our study is the first to reveal that GFRAL is expressed on human platelets and may play a significant role in the inhibitory effect of GDF-15 on platelet activation by inhibiting the AKT and ERK pathways. Thus, the GDF-15/GFRAL signaling pathway can possibly be considered a novel therapeutic target for thrombotic diseases. Certainly, our study has its limitations. Firstly, we did not directly inhibit GFRAL or delete the GFRAL gene to confirm its role in the inhibitory effect of GDF-15 on platelets. Platelets have been found to retain a subset of pre-mRNAs and can translate them into proteins [52]. As it is well known, platelets are specialized hemostatic cells characterized by the absence of the nucleus and nuclear DNA. Therefore, it is hard to inhibit GFRAL expression in platelets using RNAi. In addition, the GFRAL gene was identified in 2005, and it has not been extensively studied until recently because of its important role in the anti-obesity effect of GDF-15 [31]. Thus, there are no commercially available GFRAL inhibitors. And the use of GFRAL-knockout mice could further provide more direct evidence for the role of the GDF-15/GFRAL signaling pathway in platelets and thrombosis-related diseases. Secondly, we only investigated the effect of GDF-15 on ADP-induced platelet activation, while the effects and mechanisms of GDF-15 on platelet activation induced by other platelet agonists such as thrombin and thromboxane A2 remain to be further explored.

## Figures and Tables

**Figure 1 biomolecules-14-00038-f001:**
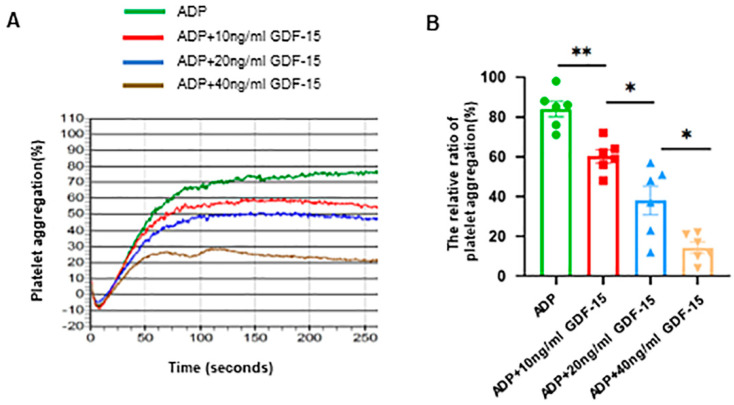
Effects of GDF-15 on human platelet aggregation. (**A**) Platelet aggregation was induced via ADP (5 μM) or ADP combined with various concentrations of GDF-15 at 10, 20, and 40 ng/mL, respectively. (**B**) The statistical analysis results of the ratio of ADP-induced platelet aggregation in humans treated with different concentrations of GDF-15 or ADP alone. *: *p* < 0.05, **: *p* < 0.01. Data were shown as the mean ± SEM, *n* = 6.

**Figure 2 biomolecules-14-00038-f002:**
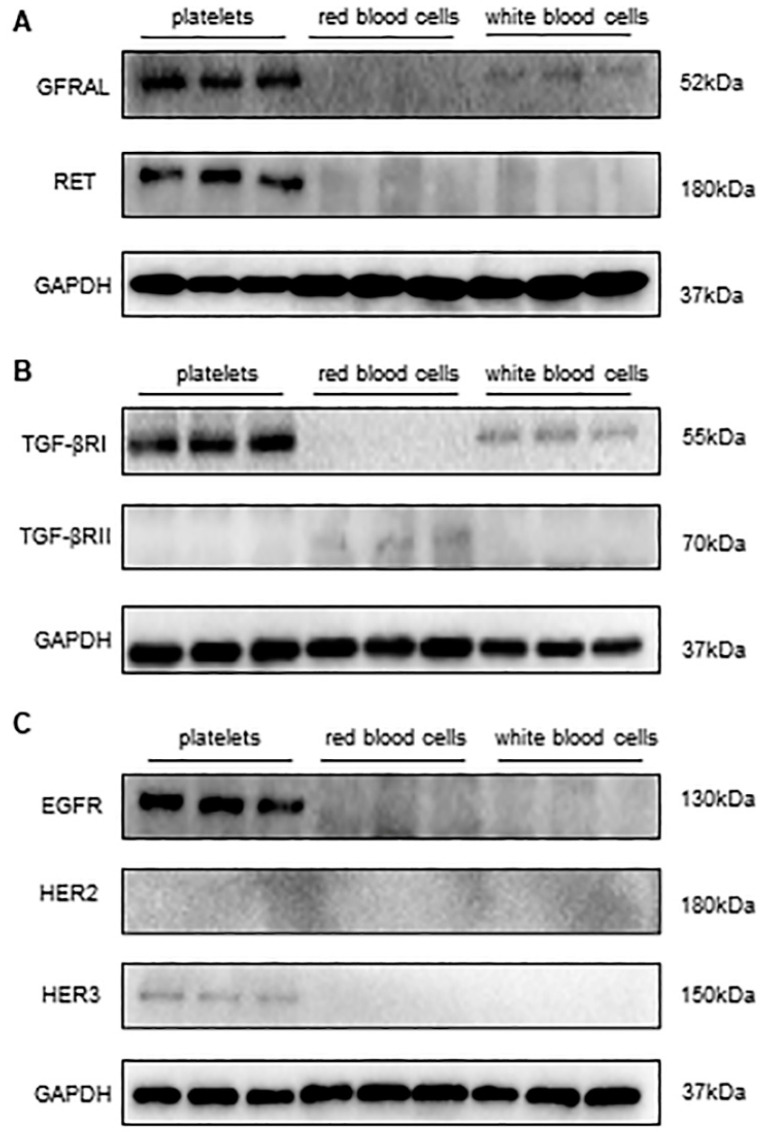
The expression of receptors associated with GDF-15 on platelets. (**A**) GFRAL and RET were detected in platelets. (**B**) TGF-βRI rather than TGF-βRII was expressed on platelets. (**C**) Among the three members of the EGFR family associated with GDF-15, EGFR, and HER3 were expressed on platelets. *n* = 6. Original images of (**A**–**C**) can be found in Appendix A.

**Figure 3 biomolecules-14-00038-f003:**
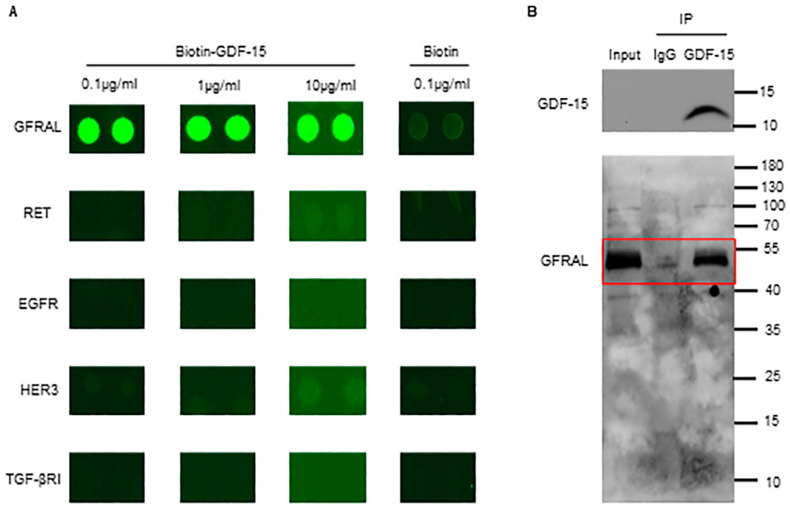
The screening for proteins that interact with GDF-15 among the receptors expressed on human platelets. (**A**) The results of the GDF-15-related receptor microarray detecting direct interactions between receptors (GFRAL, RET, EGFR, HER3, and TGF-βRI) and GDF-15 at 0.1 μg/mL, 1 μg/mL, and 10 μg/mL Biotin was used for the negative control. (**B**) The immunoprecipitation assay revealed an interaction between GDF-15 and GFRAL in human platelets. Rabbit IgG was used for isotype control. VeriBlot for the IP detection reagent, which only recognizes native (non-reduced) antibodies, was used for immunoblotting. Red box: the Western Blot band for GFRAL. Original images of (**B**) can be found in Appendix A.

**Figure 4 biomolecules-14-00038-f004:**
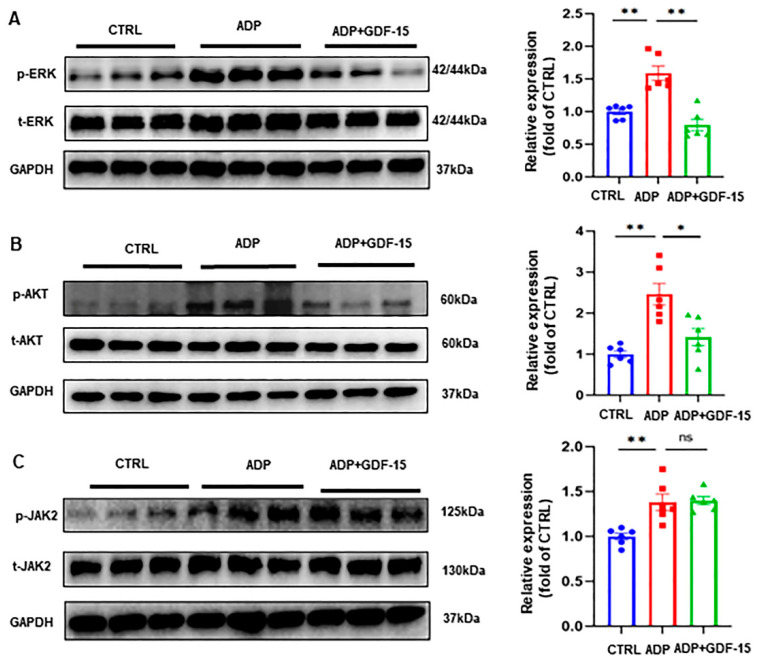
Signaling pathways associated with the effect of GDF-15 on human platelet activation. (**A**) Representative Western blot images and statistical analysis for platelet phospho-/total ERK signaling pathway. (**B**) Representative Western blot images and statistical analysis for the platelet phospho-/total AKT signaling pathway. (**C**) Representative Western blot images and statistical analysis for the platelet phospho-/total JAK2 signaling pathway. CTRL indicates no treatment was given to the platelets. Platelets were treated with 20 ng/mL GDF-15 for 15 min and incubated with 5 μM ADP for 5 min. * *p* < 0.05, ** *p* < 0.01, ns means no significance. Data are shown as the mean ± SEM, *n* = 6. Original images of (**A**–**C**) can be found in Appendix A.

**Figure 5 biomolecules-14-00038-f005:**
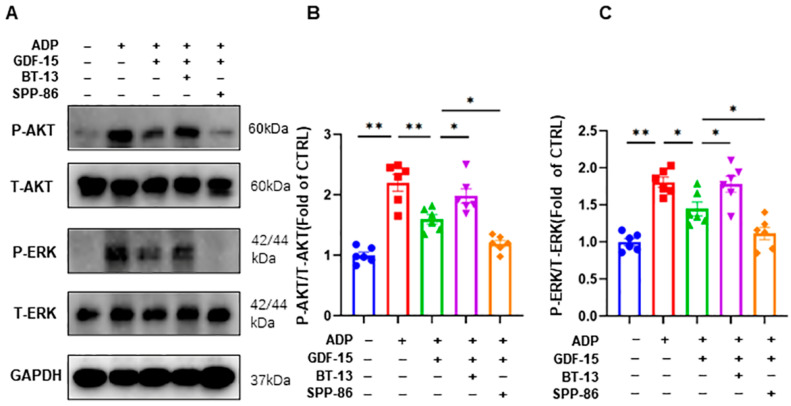
GDF-15 inhibits human platelet AKT and ERK pathways through the GFRAL/RET receptor complex. (**A**) Representative blotting images of AKT and ERK pathways in platelets after different treatments. (**B**) Statistical analysis of the relative expression level of P-AKT in platelets after different treatments. (**C**) Statistical analysis of the P-ERK relative expression level in platelets after different treatments. ADP treatment: incubated with 5 μM of ADP for 5 min; GDF-15 treatment: 40 ng/mL of GDF-15 was used for 15 min before ADP stimulation; BT-13: the RET agonist was used, pretreated with 20 μM of BT-13 for 20 min before incubation with GDF-15; SPP-86: the RET inhibitor was used, pretreated with 5 μM of SPP-86 for 20 min before incubation with GDF-15. *: *p* < 0.05, **: *p* < 0.01. Data are shown as the mean ± SEM, *n* = 6. Original images of (**A**) can be found in Appendix A.

**Figure 6 biomolecules-14-00038-f006:**
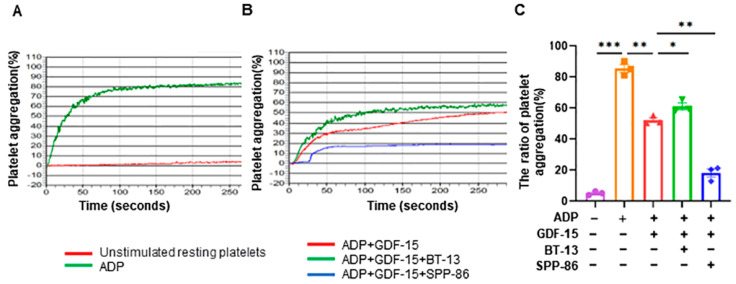
GDF-15 inhibits human platelet aggregation through the GFRAL/RET signaling complex. (**A**) Representative pictures of platelet aggregation after stimulation with or without 5 μM of ADP. (**B**) After stimulation with 5 μM of ADP and 20 ng/mL of GDF-15, the platelets were incubated with or without SPP86 or BT-13. (**C**) The statistical analysis results of the ratio of human platelet aggregation in different groups. ADP treatment: incubated with 5 μM of ADP for 5 min; GDF-15 treatment: 20 ng/mL of GDF-15 was used for 15 min before ADP stimulation; BT-13: the RET agonist was used and pretreated with 20 μM of BT-13 for 20 min before incubation with GDF-15; SPP-86: the RET inhibitor was used and pretreated with 5 μM of SPP-86 for 20 min before incubation with GDF-15. *: *p* < 0.05, **: *p* < 0.01. ***, *p* < 0.001, Data are shown as the mean ± SEM, *n* = 3.

## Data Availability

All data generated or analyzed during this study are included in this published article (and its Appendix A).

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
