# Peer review of "GDF-15 Inhibits ADP-Induced Human Platelet Aggregation through the GFRAL/RET Signaling Complex"

_biomolecules, 2023, doi:10.3390/biom14010038_

Round 1
Reviewer 1 Report (Previous Reviewer 1)
Comments and Suggestions for Authors
I have no additional comments.
Author Response
Thank you for your suggestion.
Reviewer 2 Report (Previous Reviewer 2)
Comments and Suggestions for Authors
In this original article, the authors aim to characterize the effect of GDF-15 on platelet activation induced by ADP.
1) Could the authors explain why they focused on ADP pathway in the introduction (not in the discussion) ?
2) In the introduction, the authors explain that GDF-15 is correlated with cardiovascular disease and that it can prevent the activation of platelet GPIIbIIIa which seems contradictory. The authors should comment on this in the manuscript.
3) The protocol described to isolate platelets, erythrocytes and leukocytes is unclear. It should be clearly mentionned that platelets are obtained from PRP washed twice by centrifugation at 600 g. Also this procedure is derived from a study on murine blood (ref 16) and the centrifugation speed seems not appropriate (it is suprising to pellet platelets, erythrocytes and leukocytes at the same speed)
4) The median concentration of GDF-15 even in AMI patients is around 2 ng/mL. So the authors choose concentrations 5 to 20 times higher. Could the authors comment on this ?
5) Was GDF-15 resuspended in HEPES Buffer ? If so, was the same final volume always added to PRP agregation experiments (even in the control ADP condition)? Since HEPES Buffer pH is at 6.5, it could participate in the inhibition of platelet activation.
6) Figure 1B. It is uncommon to present the results from platelet aggregation studies as ratio of ADP-induced platelet aggregation... If I understand well, the % were obtained by normalizing on ADP alone. So why aren't the 6 green points at 100 % (and not 1%...) Were they also normailed in the mean of all ADP maximal aggregation % ? This should be clearly explained.
7) Figures 5 and 6: it would have been intersting to see the effects of RET agonist (BT-13) or RET inhibitor (SPP-86) alone at the chosen concentrations.
8) Did the authors check the activation of GPIIbIIIa in their experimental conditions ?
Author Response
Thank you for your suggestion. Our reply was submitted as a Word attachment.

Round 2
Reviewer 2 Report (Previous Reviewer 2)
Comments and Suggestions for Authors
The authors answered to most of my remarks satisfactorily.
I still don't understand how they can isolate platelets at 600g and leukocytes at 2000g... This is the only point to clarify.
Author Response
Thank you for your suggestion. Our response is submitted in the form of a Word attachment.

This manuscript is a resubmission of an earlier submission. The following is a list of the peer review reports and author responses from that submission.
Round 1
Reviewer 1 Report
Comments and Suggestions for Authors
The critical role of GDF-15 in preventing agonist-induced platelet activation has been described previously. As mentioned by the authors, the indirect underlying mechanism involves a decrease in integrin activation. Here, the authors demonstrate a direct interaction between GDF-15 and the GFRAL/RET complex receptor present on the platelet surface.
Overall, the manuscript is clearly written, and the results support the conclusion.
Some questions should be addressed prior to publication:
1. The platelet aggregation studies were only performed on the basis of ADP stimulation. What is the impact of other agonists (thrombin, collagen?).
2. To verify the importance of GFRAL/RET signaling in the action of GDF-15, it would be useful to test the effect of BT13 and SPP-86 on aggregation.
3. What is the impact of GDF-15 alone on P-ERK and P-AKT? This basal condition allows the impact of ADP on GFRAL/RET signaling to be properly assessed.
4. The rationale for examining P-JAK2 is unclear. Many pathways are activated during platelet activation.
Minor comments:
5. It would be useful to know that REF functions as a co-receptor with GFRAL (and is essential for GDF-15 mediated signaling) to understand why RET agonists and inhibitors are used. This information should be included in the results.
6. The molecular weights should be indicated on the WBs.
7. Dot plots are preferable to bar plots to allow readers to appreciate the distribution of the data.
8. Positive controls for anti-TGFbetaRII and HER2 would validate the WB.
Author Response
Replies to Reviewer#1:
Our responses to Reviewer 1 was uploaded as a PDF.

Reviewer 2 Report
Comments and Suggestions for Authors
In this paper, the authors aim to study the effect of GDF-15 on platelet activation in vitro and to identify its counter receptor in platelets and the downstream signalling involved.
GDF-15 is elevated during inflammation and associated with atherosclerosis. However, the athors suggest that it inhibits platelet activation which seems contradictory... How do the authors place their results in this context?
The main critical point is that the protocol described for platelet isolation and washing in completely unappropriate. For platelet washing blood samples should be performed on ACD, not citrate, with HEPES buffer, not PBS and centrifugation steps without brake should be done at higher speed than 600g which seems insufficient to pellet platelets... At least, the authors should provide data about the purity of each cellular preparation and should also verify the absence of ex vivo platelet activation.
Finally, why did the authors chose to study only 5 uMADP induced aggregation ? What about other platelet agonists? Other platelet activation markers such as P-selectin or phosphatidylserine?
Author Response
Responses to reviewer2:
Our responses to reviewer 2 was uploaded as a Word file.

Round 2
Reviewer 1 Report
Comments and Suggestions for Authors
The authors have responded to most of my comments. It is surprising that they cannot induce platelet aggregation in response to thrombin.
Reviewer 2 Report
Comments and Suggestions for Authors
I woukd like to thank the authors to have answered honestly to my comments.
I regret that all the points were not included in the revised manuscript.
Also I understand that further experiments would need more time but I think that they would greatly improve the manuscript.
For example platelet aggregation with TRAP in PRP as the authors fail to study trombin induced aggregation...
In their reply, the authors suggest that GDF-15 inhibits ADP P2Y1 P2Y12 pathway but they don't explain how GFRAL interacts with this patway...
Finally, the cytogram of purified platelets should present CD42+ events..
In conclusion, I think that this study would beneficiate from additional work.